# Histamine Metabolism in IBD: Towards Precision Nutrition

**DOI:** 10.3390/nu17152473

**Published:** 2025-07-29

**Authors:** Dimitra Kanta, Eleftherios Katsamakas, Anna Maia Berg Gudiksen, Mahsa Jalili

**Affiliations:** Department of Nutrition, Exercise and Sports, University of Copenhagen, 1958 Frederiksberg, Denmark; xcq417@alumni.ku.dk (D.K.); rxg554@alumni.ku.dk (E.K.); trv200@alumni.ku.dk (A.M.B.G.)

**Keywords:** histamine, IBD, Crohn’s disease, ulcerative colitis, low-histamine diet, histamine intolerance, DAO, mast cell activation, histamine-producing bacteria, nutrition

## Abstract

Patients with Inflammatory Bowel Disease (IBD) exhibit a dysregulated immune response that may be further exacerbated by bioactive compounds, such as histamine. Current dietary guidelines for IBD primarily focus on symptom management and flare-up prevention, yet targeted nutritional strategies addressing histamine metabolism remain largely unexplored. This narrative review aims to summarize the existing literature on the complex interplay between IBD and histamine metabolism and propose a novel dietary framework for managing IBD progression in patients with histamine intolerance (HIT). Relevant studies were identified through a comprehensive literature search of PubMed/MEDLINE, Google Scholar, ScienceDirect, Scopus, and Web of Science. The proposed low-histamine diet (LHD) aims to reduce the overall histamine burden in the body through two primary strategies: (1) minimizing exogenous intake by limiting high-histamine and histamine-releasing foods and (2) reducing endogenous histamine production by modulating gut microbiota composition, specifically targeting histamine-producing bacteria. In parallel, identifying individuals who are histamine-intolerant and understanding the role of histamine-degrading enzymes, such as diamine oxidase (DAO) and histamine-N-methyltransferase (HNMT), are emerging as important areas of focus. Despite growing interest in the role of histamine and mast cell activation in gut inflammation, no clinical trials have investigated the effects of a low-histamine diet in IBD populations. Therefore, future research should prioritize the implementation of LHD interventions in IBD patients to evaluate their generalizability and clinical applicability.

## 1. Introduction

Inflammatory Bowel Disease (IBD) is a term that covers a heterogeneous group of chronic diseases that cause inflammation of the gastrointestinal tract (GIT), the two main ones being Crohn’s disease (CD) and ulcerative colitis (UC). Both conditions are chronic autoimmune gut diseases characterized by persistent inflammation; however, their pathological localization and clinical presentation differ [1]. The main difference is that the inflammation and/or ulcers in CD can affect any part of the GIT, while in UC they are localized only to the colon and the rectum. CD most commonly affects the final part of the small intestine, called the ileum, and/or the beginning of the large intestine, also referred to as the ascending colon, and it can involve other segments along the gastrointestinal tract [2,3]. Notably, IBD should be distinguished from irritable bowel syndrome (IBS), as IBD is a structural disease with significant long-term complications. Different classifications for IBD distinguish further conditions included in this umbrella term, but for the scope of this review, we will focus on CD and UC.

The incidence of IBD is increasing, particularly in newly industrialized countries in Eastern Europe and Asia. Historically, IBD was a disease mainly reported in developed countries, primarily Europe and North America. Still, as the prevalence of IBD has increased, it has emerged as a global phenomenon over the past two decades [4]. The prevalence of IBD varies greatly depending on the geographical region and the age group. A recent narrative review estimates that the prevalence of IBD in Europe and North America is around 187 to 832 per 100.000 and 214.9 to 478.4, respectively [5]. Notably, the age of onset of IBD typically peaks at around 15–30, and it is diagnosed due to a sudden flare-up of symptoms. Age-related classification is associated with the trajectory of the disease and includes early (childhood or very early) onset, adult onset, and elderly onset [6].

The precise etiology of IBD remains unclear; however, multiple factors contribute to its development. The primary factors contributing to the pathogenesis of IBD are genetic predisposition, environmental factors, and an abnormal, dysregulated immune response against gut microorganisms [7]. The integrity of the intestinal mucosal barrier is disrupted in IBD patients, leading to increased exposure to intestinal microbiota and luminal contents [8]. At the same time, other intestinal host defense deficiencies develop concurrently, such as reduced secretion of antimicrobial peptides by Paneth cells and abnormal mucus production resulting from a decrease in Goblet cells [9,10]. Previous studies have shown that there is an increase in proinflammatory cytokines in the inflamed mucosa of IBD patients [11,12,13]. Because the immune response is dysregulated, the inflammation remains active and becomes chronic, causing damage to the intestinal tissue and leading to various symptoms and health complications [7].

There is considerable heterogeneity in the symptomology of IBD, ranging from mild to severe, depending on the extent of the inflammation and the location in the GIT. The most common symptoms in both disorders are diarrhea, abdominal pain, rectal bleeding, extreme tiredness, and weight loss, often caused by loss of appetite [14]. Fatigue and abdominal pain are the most common presenting symptoms in CD, whereas UC patients typically present with blood-containing stools and diarrhea [15]. The diagnosis of IBD involves a combination of methods, including evaluating clinical symptoms and patients’ history, laboratory tests, physical examination, and endoscopic procedures or imaging tests [16]. IBD patients often become progressively malnourished over time, developing nutritional deficiencies, particularly iron deficiency, along with elevated energy and protein requirements. Managing malnutrition in IBD is addressed within the broader context of nutritional support for individuals with malnutrition [17].

People living with IBD experience a reduced quality of life compared to healthy individuals and have an increased likelihood of developing colon cancer, arthritis, anemia, and other health problems [18]. The main goal in treating IBD is to control the symptoms by reducing intestinal inflammation [19]. The standard pharmacological treatment includes anti-inflammatory drugs like aminosalicylates and corticosteroids to induce remission. Moreover, immunosuppressants, biological therapies, and small molecules are also used as long-term treatment therapies. If medication fails to induce remission and alleviate symptoms or if the inflammation has caused extensive damage, then doctors will recommend surgery [20].

The importance of diet and nutrition in developing and managing IBD is well known [21]. Current evidence-based dietary recommendations outlined in the clinical guidelines of ESPEN and ESPGHAN support the nutritional management of adult and pediatric IBD patients, respectively [17,22]. However, despite these advances, further progress is still needed to develop effective precision nutrition strategies to tailor dietary interventions to the individual needs of IBD patients, given the wide variability in nutritional requirements and responses.

While the primary goal of IBD treatment is to reduce inflammation and control symptoms, there is increasing interest in understanding how external factors may influence disease development and progression. Environmental exposures, dietary habits, antibiotic use, smoking, sleep, and psychological stress have all been linked to gut microbiome imbalances and disruptions in immune function, which can worsen intestinal inflammation [23]. Exploring these factors can provide valuable insights into disease management and help identify potential strategies beyond standard medication. One such factor is histamine, a compound involved in immune responses and gut function [24]. Histamine has been shown to play a role in gastrointestinal inflammation, and both direct and indirect pathways of histamine activity have been implicated in the development of IBD. This suggests that altered histamine metabolism may contribute to disease severity, making it a potential target for future treatment approaches [25]. This narrative review aims to provide an overview of histamine’s molecular signaling pathways involved in IBD and to evaluate the potential therapeutic role of a low-histamine diet in a subset of IBD patients.

## 2. Methods

A narrative literature search was conducted across PubMed/MEDLINE, Google Scholar, ScienceDirect, Scopus, and Web of Science using combinations of MeSH terms and free-text keywords relevant to histamine signaling, immune pathways, and dietary strategies in IBD. The search included English-language articles published over the past 10–15 years, with exceptions made for foundational studies relevant to the topic. Both original research articles in humans and animal studies, systematic reviews, and review papers were included based on their relevance to histamine signaling, IBD, and dietary interventions. A complete list of search terms and combinations used is provided in Appendix A.

## 3. The Complex Interplay Between Histamine and IBD

### 3.1. Endogenous Histamine Dynamics: Production, Degradation, and Enzyme Dysregulation in IBD

Histamine is a bioactive amine synthesized via the decarboxylation of histidine by L-histidine decarboxylase (HDC). It is primarily stored in mast cells and basophils, where it is packaged into secretory granules by the Golgi apparatus [26]. The release of histamine and cell degranulation occurs when a specific antigen binds to FcεRI (high-affinity IgE receptor), triggering protein kinase C (PKC)-mediated synthesis of pro-inflammatory lipid mediators and cytokines. Histamine plays a role in various physiological processes, including regulating cardiovascular function, modulating gastrointestinal activity by stimulating mucus and gastric acid secretion, and involvement in neurotransmission. The degradation of histamine is primarily mediated by two enzymes: diamine oxidase (DAO) and histamine-N-methyltransferase (HNMT). DAO is responsible for degrading extracellular histamine, while HNMT metabolizes intracellular histamine. The breakdown products are imidazole acetaldehyde from DAO activity and N-methylimidazole acetic acid from HNMT activity. DAO deficiency, which can have genetic, pharmacological, or pathological origins, disrupts histamine homeostasis and increases histamine absorption [27] (Figure 1).

In IBD, it has been observed that reduced DAO activity contributes to impaired histamine metabolism and exacerbated symptoms. The chronic and acute inflammation associated with IBD often leads to significant villous atrophy, particularly affecting the villous tips [28,29]. DAO, the key enzyme responsible for histamine degradation, is predominantly localized in the mature villous cells at the apical ends of the small intestine [30]. Structural damage to these regions results in diminished DAO production, consequently impairing histamine degradation within the GIT.

Excess histamine may further exacerbate compromised intestinal barrier integrity and inflammation by allowing increased histamine permeation into surrounding tissues and the circulatory system. This cascade of events could contribute to systemic symptoms beyond the primary gastrointestinal manifestations of IBD. Indeed, an increase in mucosal histamine content and histamine secretion has been observed in both CD and UC [26]. Additionally, DAO production is highest in the small intestine, peaking in the ileum, the final section of the small intestine, which coincides with ileal CD as the most common form of Crohn’s; approximately one-third of patients present with localized inflammation in this region [27]. A diet high in histamine may oversaturate the remaining DAO in this area, leading to excess histamine diffusing into the compromised tissue and further exacerbating surrounding tissues. Although histamine is endogenously synthesized by immune cells, such as mast cells and basophils, exogenous sources include the gut microbiome [26] and histamine-containing foods [31].

A decreased capacity for histamine breakdown can lead to histamine intolerance (HIT), characterized by the accumulation of histamine in the body. HIT is considered a non-allergic food hypersensitivity reaction, with a wide range of clinical manifestations due to the presence of four distinct histamine receptors [32,33]. These symptoms are often nonspecific and may affect both gastrointestinal and extraintestinal systems. In a study with 133 histamine-intolerant patients, the most frequent symptoms were abdominal distension, fullness, diarrhea, abdominal pain, and constipation [34].

Studies on DAO genetic variability found more than 50 nonsynonymous single-nucleotide polymorphisms (SNPs) of the DAO-encoding gene [35,36,37]. Most relevant DAO genes affect Caucasian individuals, but others explain the enzyme deficiency of Asian or African individuals [38]. Given that DAO deficiency has a genetic basis, one possible diagnostic approach involves identifying SNPs that indicate a population’s susceptibility to histamine.

### 3.2. Histamine Receptors and Immune Modulation

Histamine’s effects on the immune system are context-dependent. Histamine can exert both pro-inflammatory and anti-inflammatory effects, depending on the type of histamine receptors (HRs) it activates and the downstream immunological signaling pathway involved. In the GIT, research suggests that HR signaling involves complex interactions between immune and epithelial cells, influencing inflammatory outcomes in various conditions [39]. H1R and H4R are associated with pro-inflammatory responses, while H2R has been shown to mediate anti-inflammatory effects. All HRs are expressed in the GIT (with H4R mRNA being less abundant), except for H3R, which remains a topic of debate and requires further research [40,41] (Table 1).

Although histamine plays a role as a mediator of inflammation, the underlying mechanisms are underexplored. However, histamine has been implicated in chronic inflammation in response to allergies, inflammation, elevated histamine levels from food intake, impaired activity of the DAO and HNMT enzymes, and microbial histamine metabolism in the gut [26]. Increased histamine levels can disrupt intestinal homeostasis, potentially leading to HIT or histamine-related symptoms [42]. This occurs through histamine’s ability to sensitize nerve endings and modulate pain signaling, manifesting as visceral hypersensitivity (VH) in the intestine. Moreover, H1R and H4R antagonists have been shown to reduce post-inflammatory VH dose-dependently [43]. While the detailed mechanisms remain unknown, it is hypothesized that histamine signaling contributes to chronic inflammation and VH in IBD patients. The expression and functional activity of HRs have also been altered in IBD, possibly due to the high inflammatory state in the gut [44,45]. The strong immunomodulatory effects of histamine and the complex immune-mediated mechanisms underlying IBD underscore its importance in disease development. However, further research is needed to clarify these mechanisms and evaluate histamine signaling as a therapeutic target.

Mast cells detect harmful stimuli and initiate immune responses by releasing a range of bioactive mediators, including histamine, cytokines, chemokines, and proteases, highlighting their role in host defense and mucosal immunity. These mediators contribute to vascular modulation, immune cell recruitment, and the regulation of inflammation. In IBD, mast cells contribute to disease pathogenesis, which are found in increased numbers in the intestinal mucosa of IBD patients, particularly in the ileum and the colon [46]. Early studies demonstrated that mast cells from IBD colonic tissue released significantly more histamine than healthy controls. In dextran sulfate sodium (DSS)-induced colitis Ws/Ws rat models, mast-cell-deficient rats exhibited altered disease severity, suggesting that mast cells modulate disease severity [47]. In the acute phase of DSS colitis, reduced colonic histamine levels, despite elevated systemic concentrations, support localized histamine release and its potential involvement in mucosal damage. Notably, increases in mast cell numbers and circulating histamine may result in the elevation of neuropeptides, such as somatostatin, suggesting a role in early inflammatory signaling [48].

Mast cell activation, primarily through IgE/FcεRI-mediated degranulation, contributes to intestinal barrier dysfunction and increased permeability [49]. This promotes antigen translocation into the mucosa, leading to further activation of mast cells and CD4^+^ T cells, amplifying the inflammatory cascade [50]. The co-occurrence of mast cells and macrophages in the lamina propria may reflect intercellular signaling contributing to chronic inflammation. In parallel, mast cells secrete both proinflammatory and regulatory cytokines, including IL-2 and IL-10, which play critical roles in modulating immune responses in the gut [51]. Histamine released from mast cells influences this cytokine balance via receptor-specific mechanisms; activation of H_1_ receptors favors IL-2 production and Th1 polarization, while H_2_ receptor signaling enhances IL-10 secretion and exerts anti-inflammatory effects [52].

In histamine-deficient mice (HDC^−^/^−^) subjected to DSS-induced colitis, a marked reduction in IL-10–producing immune cells has been observed [53], underscoring the immunomodulatory role of mast-cell-derived histamine in intestinal inflammation. Furthermore, tryptase release from mast cells, a serine protease implicated in tissue remodeling and fibrogenesis, positions mast cells as potential mediators in the development of intestinal fibrosis, a long-term complication of chronic IBD [54].

## 4. Endogenous Microbial Factors: Microbial Dysbiosis and Histamine-Producing Bacteria

### 4.1. Microbial Dysbiosis in IBD

In IBD, this dysbiosis is characterized by a decline in Bacteroidetes and Firmicutes, including the beneficial butyrate-producing bacteria *Faecalibacterium prausnitzii* and *Roseburia* spp., and an increase in *Proteobacteria*, many of which are opportunistic [23,55,56]. This imbalance leads to mucosal barrier dysfunction, exposing the submucosa to bacteria, fueling a cycle of inflammation and tissue damage [57]. One mechanistic link may be via altered metabolites, such as decreased short-chain fatty acids (SCFAs), which can dysregulate epithelial turnover by activating Toll-like receptors (TLRs) on intestinal cells [58]. Several factors, including transit time, pH, diet, and antibiotic use, influence the gut microbiota and are relevant to the development of IBD [59,60,61].

During fiber fermentation in the colon, certain gut bacteria produce histamine and other metabolites that influence local gut function. In disease flare-ups, alterations in colonic transit time (CTT) may disrupt this microbial activity, including the conversion of primary to secondary bile acids, potentially contributing to impaired bile acid metabolism [62]. In cases where IBD patients experience constipation (slower CTT), the accumulation of secondary bile acids may exert toxic effects on colonic cells by inducing DNA damage [63]. Prolonged transit time and altered pH, particularly in the colon, are associated with a shift from saccharolytic to proteolytic fermentation, leading to the accumulation of branched-chain fatty acids (BCFAs), phenols, indoles, ammonium (NH_3_), and hydrogen sulfide (H_2_S) [64]. Both H_2_S and NH_3_ are implicated in IBD pathogenesis, as they can exacerbate mucosal inflammation and compromise epithelial barrier integrity. These observations highlight the relevance of personalized nutritional and therapeutic strategies that consider both host and environmental factors influencing gut microbial dynamics.

### 4.2. Histamine-Producing Bacteria in IBD

Although research highlights the role of endogenous histamine production by host immune cells in IBD, the microbial contribution remains underexplored. Certain bacteria can produce histamine by converting the amino acid histidine to histamine through the enzyme histidine decarboxylase [65]. Two major families of histidine decarboxylases (HDCs) have been identified: pyridoxal-5′-phosphate (PLP)-dependent HDCs, observed in Gram-positive bacteria, which require PLP as a cofactor to produce histamine and CO_2_, and pyruvyl-dependent HDCs observed in Gram-negative bacteria, which instead utilize a covalently bound pyruvyl moiety [66]. Histamine-producing bacteria typically harbor hdc operons, which are co-regulated gene clusters that enable histamine synthesis and export in the bacterial cytoplasm. The activity of the histidine/histamine antiporter HdcP is essential for importing histidine and exporting histamine. The gene encoding HDC is hdcA, and it is usually found within an operon alongside hdcB, hdcP, and the transcriptional regulator hdcR [67] (Table 2).

Various factors, including histidine availability, histamine levels, and environmental pH, modulate the expression of HDC genes [68]. Elevated histidine concentrations can induce HDC gene expression, whereas histamine has been shown to repress it in several lactic acid bacterial strains [69]. Regarding pH, histamine production may enhance bacterial survival under acidic conditions by helping maintain a neutral cytosolic pH, as the decarboxylation of histidine releases CO_2_ and H^+^. In *Lactobacillus vaginalis*, the HDC pathway serves as an acid stress resistance mechanism, supporting its survival in low-pH environments, such as during dairy fermentation [70,71]. Similarly, pathogenic bacteria like *Escherichia coli* utilize alternative amino acid decarboxylases to resist acid stress during passage through the stomach [72].

Despite the potential clinical relevance of histamine in gastrointestinal disorders, such as IBD, HIT, or small intestinal bacterial overgrowth (SIBO), there remains a gap in understanding which bacteria can produce and secrete histamine. A study involving individuals with HIT found a significant imbalance in gut microbiota compared to healthy controls, characterized by a reduced presence of beneficial bacteria and an increased abundance of histamine-producing species [73]. While HIT and IBD are distinct conditions, this shared pattern of dysbiosis and elevated histamine-producing bacteria suggests a possible overlapping mechanism, where altered microbial activity may contribute to histamine sensitivity and immune dysregulation in IBD. Little is known about the histamine-producing taxa enriched during dysbiosis that carry the *hdc* operon, but several have been identified in both environmental and clinical contexts. These include specific strains of *Lactobacillus reuteri*, *Morganella morganii*, *Enterobacter cloacae*, and other members of the *Enterobacteriaceae* family [65]. For example, *M. morganii* has been consistently identified as a potent histamine producer, with expression of the *hdc* operon leading to significant histamine accumulation in food and in vivo systems [31]. Likewise, *L. reuteri* expresses a PLP-dependent histidine decarboxylase encoded by *hdcA*, which is co-regulated with *hdcP*, *hdcB*, and *hdcR*, facilitating histamine production under acidic conditions [74]. Metagenomic analyses of colonic samples from individuals with IBD have further revealed an enrichment of Enterobacteriaceae members with decarboxylase gene potential, suggesting that microbial-derived histamine may contribute to mucosal immune dysregulation and symptom exacerbation in IBD [75] (Figure 2).

## 5. Towards Precision Nutrition in IBD: Low-Histamine Diet and Beyond

### 5.1. Modulating Exogenous Histamine Levels: Low-Histamine Diet

A low-histamine diet is the gold standard for HIT treatment [39,76], and, given the complex role of histamine in inflammatory processes, we suggest that IBD patients exhibiting symptoms and biomarkers, such as serum DAO, indicative of HIT, may benefit from its implementation [77,78]. Reducing histamine intake in these high-risk individuals could help modulate endogenous histamine-driven pro-inflammatory mechanisms and support better disease management. There are multiple mechanisms proposed as to why IBD patients may exhibit histamine-related symptoms [26]. As mentioned, decreased DAO activity in IBD might result in reduced histamine degradation and, thus, higher histamine levels accumulating in the body [79]. Furthermore, endogenous release of histamine from the granulocytes of mast cells is heightened in IBD patients due to an altered phenotype in the inflamed tissues [80].

Based on current evidence, minimizing the exogenous intake of histamine may be beneficial for people with IBD. Although there is no consensus on specific foods that should be excluded from a low-histamine diet, some food groups, such as fermented foods, are consistently identified in the literature [81]. These high-histamine foods (e.g., aged cheese, fermented meats, or stale fish) contain elevated histamine due to bacterial fermentation, whereas decarboxylation of amino acids occurs during aging or spoilage. The consumption of high-histamine foods can directly increase histamine levels in the gut.

Besides foods with high levels of biogenic amines, certain foods may act as histamine-releasing agents, triggering mast cells to release histamine even when they contain low histamine levels. Others may competitively inhibit the DAO activity and reduce histamine degradation, which is known to primarily occur with the presence of putrescine or cadaverine in foods [33,81].

Food groups are therefore classified based on whether they increase histamine load directly or interfere with its metabolism (Table 3). Despite the limited research and variations in histamine food content across regions, this evidence-based classification helps explain why excluding certain foods may benefit IBD patients. These foods are either high in histamine levels or impair the body’s ability to handle histamine. Therefore, a general recommendation is to avoid processed, fermented, or aged foods, as well as histamine-releasing foods, and to prefer fresh, unprocessed foods, which tend to have lower histamine levels and minimize exposure.

### 5.2. Modulation of Endogenous Histamine-Producing Bacteria

Given the significant role of histamine in various physiological and pathological processes, modulating its production by altering the gut microbiota represents an attractive therapeutic approach. A recent study by Chen et al. analyzed longitudinal stool samples from the Integrative Human Microbiome Project 2 (iHMP) involving IBD patients to identify bacteria potentially linked to histamine production in their gut microbiome [87]. There is an increased abundance of histamine in IBD patients and genes encoding histidine decarboxylases in the microbiome of patients with CD. Specific bacterial species and strains have been identified as histamine-producing. All of the strains of bacterial species *M. morganii* and specific strains of *L. reuteri* and *Enterobacteriaceae* have been associated with increased histamine levels [87,88,89].

Multiple strategies have been proposed to modulate bacterial histamine production. Reducing the dietary intake of histamine-rich foods decreases the availability of histidine and thus histamine production by specific bacteria [90]. Probiotics or prebiotics can also shift microbial communities by promoting the growth of specific bacteria that compete with histamine-degrading bacteria [91]. As a pharmacological approach, the blockade of the receptor H4R has been proposed as a method to reduce mast cell accumulation and visceral hyperalgesia in a cohort of IBS patients [92]. H4 receptor antihistamines have shown preclinical efficacy in colitis and atopic dermatitis [93,94]. It remains unclear whether these bacteria contribute to the excess of local histamine in IBD, whether microbial-derived histamine interacts with host receptors to modulate inflammation, and whether bacterial histamine can initiate inflammation, amplify ongoing immune responses, or interact with host histamine receptors in a concentration-dependent manner.

### 5.3. Role of Fiber and Histamine

Dietary fiber is widely recognized for its beneficial effects on gut health, primarily through fermentation by gut microbiota into short-chain fatty acids (SCFAs), such as butyrate, acetate, and propionate, which exert anti-inflammatory effects and support epithelial integrity [95]. Currently, IBD dietary guidelines often recommend limiting the intake of dietary fiber during flare-ups and gradually reintroducing it during remission [17]. In IBD patients, a combination of reduced fiber intake and microbial dysbiosis can elevate colonic pH and shift microbial metabolism from saccharolytic to proteolytic fermentation [62]. This metabolic shift leads to increased luminal concentrations of free amino acids, such as histidine, which can be decarboxylated into histamine by specific gut microbes [96]. Among these, *Morganella morganii*, a known histamine-producing bacterium found in higher abundance in IBD patients, especially UC patients [97], may benefit from the increased availability of amino acids and a more alkaline colonic environment, conditions that favor its growth and enzymatic activity [98].

### 5.4. Role of Yeast and Salt in IBD: Links to Histamine Metabolism

Yeast [99] and salt [100] are food-derived components that have been associated with the development and exacerbation of CD. Elevated levels of IgG and IgA antibodies against *Saccharomyces cerevisiae*, known as anti-*Saccharomyces cerevisiae* antibodies (ASCA), have been observed in various autoimmune conditions, including CD, and serve as serological markers of immune dysregulation [101,102,103,104,105]. Importantly, dietary yeast and yeast-containing fermented foods can also be significant sources of exogenous histamine and other biogenic amines, which may contribute to histamine accumulation in the gut [106].

A recent study in *Nature Medicine* demonstrated that food-derived yeast can directly activate dysregulated CD4^+^ T-cell responses in CD patients [107], contributing to intestinal inflammation. The study utilized peripheral blood mononuclear cells (PBMCs) from IBD patients and healthy donors, stimulating them with yeast lysate and measuring ASCA levels. The findings confirmed a heightened immune response in CD patients. Given the pivotal role of CD4^+^ T cells in CD pathology, these findings align with prior studies highlighting oligoclonal expansions of NKG2D-expressing CD4^+^ T cells in the disease [108]. Moreover, an older interventional study examining yeast exclusion in 19 patients with CD reported a significant reduction in the mean Crohn’s Disease Activity Index (CDAI) during the exclusion period [109]. Furthermore, the Crohn’s Disease Exclusion Diet (CDED), which, among others, eliminates yeast, has also demonstrated clinical efficacy in a recent study on pediatric and adult populations [110].

Additionally, an increasing amount of evidence suggests that excessive salt intake, common in Western diets, may contribute to immune dysregulation and is associated with the development of IBD and other autoimmune diseases [100]. Although there is no precise mechanism for how salt intake is involved in the etiopathogenesis of IBD [111], preclinical data support its potential pro-inflammatory role, which may include effects on histamine metabolism. Exclusive Enteral Nutrition (EEN) is effective in inducing remission in CD, and emerging evidence suggests that lower dietary salt intake may contribute to its anti-inflammatory effects [112].

Increased dietary sodium intake can alter the gut microbiota’s composition, decreasing certain beneficial species, such as certain Lactobacillus species, which play a role in degrading dietary histamine, and promoting the expansion of pro-inflammatory TH17 cells [113]. Another study showed that a high-salt diet can exacerbate colitis symptoms in mice by reducing the relative abundance of *Lactobacillus* species and decreasing butyrate production in the colon [114]. This effect is likely mediated through p38/MAPK signaling pathway alterations. Other studies have shown that excess sodium can worsen intestinal inflammation by boosting the production of pro-inflammatory cytokines like IL-17A and TNF-α in gut immune cells and may even lead to sodium buildup in the colonic tissue itself, which could further disrupt the epithelial barrier and amplify inflammation [111,115]. Importantly, both yeast and salt may converge on a shared histamine-related pathway. CD4^+^ T-cell activation by dietary yeast can stimulate mast cell degranulation and subsequent histamine release, which is already heightened in IBD mucosa [107]. At the same time, high salt intake reduces certain *Lactobacillus* species, some of which are associated with gut homeostasis and might indirectly favor the dominance of histamine-producing species [98]. Moreover, many high-salt processed foods, such as aged cheeses and cured meats, are also rich in histamine, further compounding the overall histamine burden [116]. Considering their role in immune activation and histamine accumulation, moderating yeast and salt intake may be a practical dietary consideration for managing IBD.

## 6. Discussion

This review proposes an additional dietary intervention focused on reducing histamine-containing foods and modulating microbiome composition as a novel approach for managing various stages of IBD progression in patients with HIT or mast cell activation. Importantly, this approach is intended as a complementary strategy and should not replace established medical treatments for IBD. A low-histamine diet excludes foods with high histamine concentrations and histamine-releasing foods. A more detailed overview of LHD involves excluding fermented and processed food products and limiting the intake of yeast and salt as factors that promote the activity of histamine-producing bacteria. Fiber should be avoided only in flare-up states, aligning with the current guidelines for IBD management. Its gradual reintroduction during remission may be aligned with an individual’s symptoms to support mucosal healing.

Current dietary guidelines for IBD focus on managing symptoms, supporting remission, and reducing inflammation. Multiple diets have been studied, including low FODMAP, Mediterranean, vegetarian, and specific carbohydrate diets (low in grains, sugars, and lactose). In a randomized controlled trial comparing the Mediterranean diet with a specific carbohydrate in adults with CD, the study failed to show a superior diet for patients with mild to moderate symptoms [117]. The low FODMAP diet has been shown to improve quality of life and gastrointestinal symptoms, but it is not ideal to alleviate mucosal inflammation [118]. For example, in IBD patients with IBS criteria in remission, a low FODMAP diet for 6 weeks improved gut symptoms and diarrhea but not constipation or inflammatory markers [119]. Future guidelines should prioritize the development of personalized dietary strategies tailored to specific subgroups, focusing on mitigating inflammation to support long-term disease regression rather than solely symptom management.

The risk of malnutrition should be considered in any dietary intervention, including implementing LHD. Notably, patients with IBD are considered a high-risk population for malnutrition [17]. There are concerns regarding diets low in fermentable saccharides and polyols and their long-term effects on gut microbiota and nutritional adequacy [120]. In LHD, the exclusion of fermented foods, which are rich in probiotics, may be replaced by probiotic supplementation. Aged meats, fermented foods, and certain dairy products are rich sources of iron and vitamin B_12_, micronutrients frequently deficient in individuals with IBD [121]. Similarly, shellfish provide zinc, another essential nutrient often found to be reduced in this population [122].

To advance towards personalized nutrition in IBD, it is essential to establish criteria for identifying patients who exhibit symptoms consistent with HIT and show hypersensitivity to elevated histamine levels. In this subgroup of patients, excluding histamine may facilitate digestive recovery and promote intestinal epithelial remission. Measuring histamine levels is challenging due to their rapid fluctuations and short half-life. While IgE is used as a biomarker of disease activity in IBD, allergy testing in HIT shows no signs of IgE-mediated food allergy in allergy tests. Studies measuring IgE levels in IBD patients have yielded mixed results [123,124], underscoring the need for personalized approaches to assessment and management.

Recent research has identified a positive association between ultra-processed food (UPF) intake and urinary 1-methylhistamine (MHA), a potential biomarker for HIT [125,126]. MHA is produced via the methylation of histamine by HNMT and may serve as an alternative to serum DAO activity in diagnosing HIT. The underlying mechanisms linking UPF to elevated MHA remain unclear but may involve food processing techniques or additives. Advances in metabolomics, including the use of ultra-high-performance liquid chromatography (UHPLC), have improved the accurate quantification of histamine and MHA in biological samples [127].

A key area of interest lies in the altered activity of host histamine-degrading enzymes, DAO and HNMT, which are responsible for degrading extracellular and intracellular histamine, respectively. Alterations in the activity or expression of these enzymes can contribute to histamine accumulation and symptom manifestation. DAO, primarily expressed in the small intestine, plays a particularly critical role in intestinal histamine metabolism, and its deficiency has been observed in subsets of patients with gastrointestinal disorders, including IBD. The measurement of DAO enzymatic activity is feasible through colorimetric techniques [128] or a zymographic approach [129] or spectrophotometry. However, DAO quantification levels remain challenging. According to an available commercial kit, cut-off values < 3 U/mL are considered decreased, 3–10 U/mL slightly decreased, and ≥10 U/mL normal [130]. Thus, a diagnostic approach similar to that used in HIT patients could be applied [131]. Combining a histamine-restricted diet with structured dietary reintroduction in suspected hypersensitivity among IBD patients can be conducted alongside assessment of DAO level measurements. However, current DAO activity assays do not always reflect tissue-level activity, particularly in inflamed intestinal regions, limiting their diagnostic reliability. Additionally, there is a need to assess allergic symptoms, as histamine and mast cell behavior in IBD closely resemble that observed in certain allergic disorders, highlighting potential shared pathophysiological mechanisms [132].

Genetic polymorphisms in DAO and HNMT genes may impact enzymatic efficiency, with some variants linked to reduced enzyme expression or activity. Although the DAO SNP rs1049793 does not appear to influence genetic susceptibility to CD [133], its role in disease phenotype, symptom severity, or response to histamine-restricted diets remains to be clarified. Similarly, HNMT activity, though less studied in the context of IBD, may also contribute to histamine imbalance, especially in extra-intestinal manifestations.

Although gut dysbiosis is well-documented in IBD, the specific contribution of histamine-producing bacteria, which are enriched in IBD patients, as observed in 2451 fecal metagenome analyses [65], remains poorly understood. The enrichment of histamine-producing bacteria appears to differ across patient cohorts and is not consistently associated with any single bacterial phylum. For instance, some studies identified increased levels of *Actinobacteriota* and Firmicutes [134], while others reported enrichment of Firmicutes and Proteobacteria [135]. In the HMP2 dataset, *Bacteroidiota* and Proteobacteria were most prevalent, whereas an increase in *Actinobacteriota* was reported elsewhere [136]. Despite this variability, all studies reported a higher abundance of HDC gene clusters, the key enzymes for microbial histamine synthesis, in IBD samples.

## 7. Limitations

This review summarizes the existing literature on the relationship between IBD and histamine and evaluates the potential therapeutic application of a low-histamine diet. Several limitations affect the generalizability and clinical applicability of the LHD approach in patients with IBD. To date, no clinical trials have investigated the effects of a low-histamine diet in IBD populations, despite growing interest in the role of histamine and mast cell activation in gut inflammation. Additionally, the practical implementation of an LHD is challenging due to variability in histamine content across foods, influenced by factors like processing methods and regional differences. Given the high interindividual variability in disease presentation among patients, responses to dietary interventions are expected to be highly individualized, underscoring the importance of personalized nutrition rather than a one-size-fits-all approach. Although alternatives for low-histamine foods have been proposed (Table 3), the restrictive nature of the diet raises concerns regarding potential nutritional inadequacies. Individual differences in genetic background, enzyme activity, and microbiome composition further complicate dietary recommendations. Finally, due to their rapid fluctuations, methodological challenges in reliably measuring histamine levels limit the establishment of clear clinical correlations between dietary histamine intake and symptom severity in IBD patients.

## 8. Future Perspectives

Given the multifactorial nature of IBD, with contributing roles from genetics, nutrition, microbiota, immune dysregulation, and environmental factors, future research must move beyond traditional study design models that test one intervention at a time. While RCTs remain the gold standard for establishing causality, traditional designs may not fully capture the complex, dynamic interactions that characterize chronic inflammatory diseases. Innovative approaches, such as multi-component interventions and adaptive trial designs, including omics analyses, should be considered to capture heterogeneity between IBD patient subgroups.

Future clinical studies could explore phase alterations, implementing dietary transitions with an LHD to minimize adverse reactions in histamine-sensitive individuals. For instance, short-term restriction of fermentable carbohydrates (e.g., low FODMAP principles) followed by gradual reintroduction of soluble fiber may help modulate microbial fermentation and support gut barrier integrity. Although soluble fiber is generally beneficial for mucosal healing and short-chain fatty acid production, its rapid fermentation can produce osmotic stress and stimulate histamine production, particularly in individuals with dysbiosis and impaired intestinal permeability. The use of probiotic supplementation in these cases could be beneficial in replacing the benefits of fermented foods in LHD or fermentable foods like fiber in sensitive individuals.

Furthermore, microbial overproduction of histamine may saturate the host’s enzymatic degradation capacity, especially if DAO activity is compromised. This overload could contribute to systemic symptoms, mucosal damage, or nutrient malabsorption. Future research should investigate these microbial–host interactions using metagenomics, metabolomics, and functional enzyme assays to clarify whether microbial histamine load contributes to symptom flares or weight loss in IBD patients. Additionally, emerging evidence suggests that the gut virome may play a significant role in shaping gut microbial ecology and host metabolism, including potential impacts on histamine metabolism [137,138]. Future studies should explore the contributions of the gut virome to IBD pathophysiology and investigate how dietary interventions could modulate virome composition and function in precision nutrition strategies.

Ultimately, identifying histamine-sensitive IBD phenotypes through biomarkers, such as DAO activity, genetic polymorphisms, or microbial histidine decarboxylase gene abundance, could allow for the development of more personalized dietary interventions and therapeutic targets.

## Figures and Tables

**Figure 1 nutrients-17-02473-f001:**
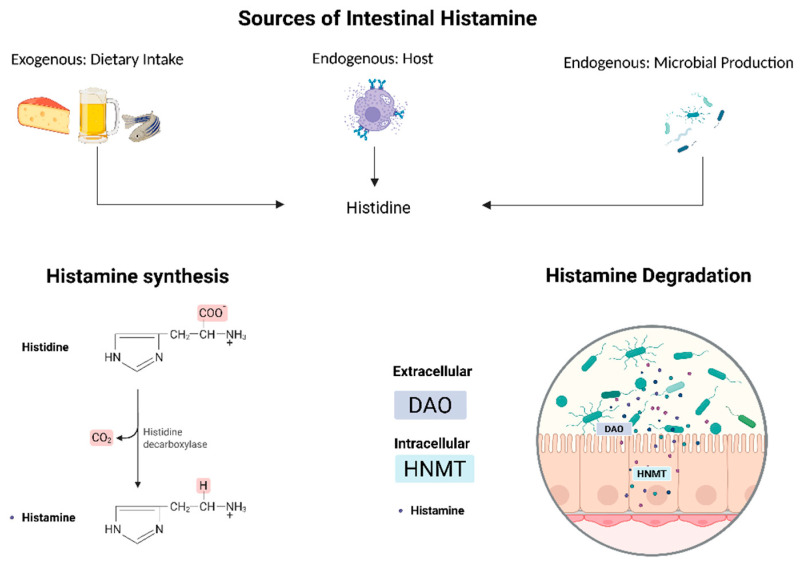
Histamine’s endogenous and exogenous sources, synthesis, and degradation enzymes intra- and extracellularly. Abbreviations: diamine oxidase (DAO); histamine-N-methyltransferase (HNMT).

**Figure 2 nutrients-17-02473-f002:**
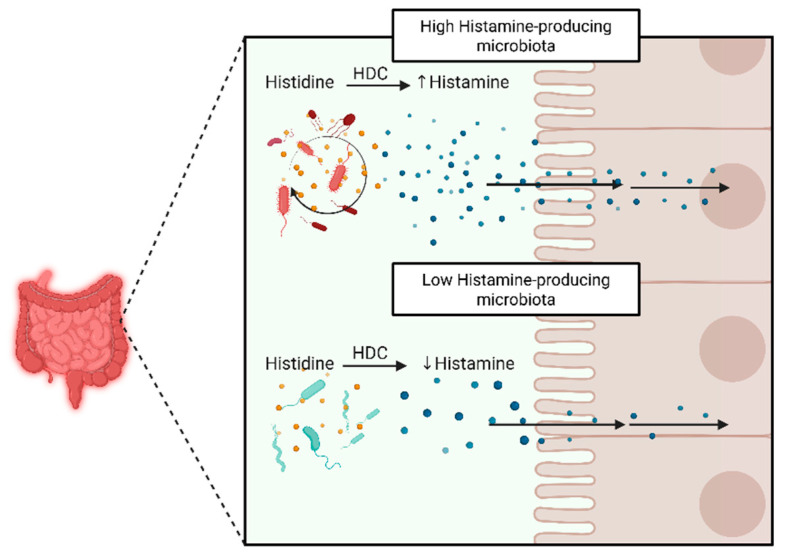
Histamine-producing bacteria’s metabolism and gut epithelial diffusion. Abbreviations: histidine decarboxylase (HDC).

**Table 1 nutrients-17-02473-t001:** Histamine receptors and potential roles in IBD [26,39].

Receptor	G Protein Coupling	Expression	Molecular Mass (kDa)	Potential Role in IBD
H1R	Gαq	Enterocytes	56	Development of allergic reactions
H2R	Gαs	Enterocytes Gastric pancreatic cells	40	External secretion of hydrochloric acid Anti-inflammatory effects Innate immune response to microorganisms when histamine binds
H3R	Gi/o	Nervous system: hippocampus, cerebral cortex, and neurons of the basal ganglia	48	Pro-inflammatory activity
H4R	Gi/o	Small and large intestinal epithelium, bile, pancreatic duct	44	Mainly present in immune cells May contribute to the development of inflammatory reactions and hypersensitivity

Abbreviations: histamine receptor (HR), G protein alpha q subunit (Gαq), G protein alpha s subunit (Gαs), G protein alpha i and o subunits (Gi/o), kilodalton (kDa).

**Table 2 nutrients-17-02473-t002:** Overview of the hdc operon genes, functions present in histamine-producing bacteria [66,67].

Gene	Function	Role
hdcA	Histidine decarboxylase	Catalyzes the decarboxylation of histidine to histamine (PLP-dependent)
hdcP	Histidine/histamine antiporter	Imports histidine Exports histamine from the bacterial cytoplasm
hdcB	Maturation Protein	
hdcR	Transcriptional regulator	Part of the LysR-type transcriptional regulator (LTTR) regulating amino acid metabolism pathways

**Table 3 nutrients-17-02473-t003:** Classification of foods based on their impact on histamine load.

Low-Histamine	High-Histamine	Histamine-Releasing
Fresh dairy (ricotta, mozzarella, cottage cheese, milk)	Aged/fermented cheeses (parmesan, cheddar, blue cheese)	Citrus and tropical fruits (orange, lemon, pineapple, banana, avocado)
Fresh meat and fish (chicken, beef, trout, cod, etc.)	Fermented foods and drinks (kimchi, yogurt, kombucha, wine, beer)	Certain vegetables (tomato, eggplant, spinach, squash)
Soy and meat alternatives (e.g., coconut aminos)	Processed meats (bacon, salami, sausages)	Berries and chocolate (strawberries, raspberries, cocoa)
Unflavored distilled alcohol (vodka, gin)	Preserved fish (canned tuna, sardines, smoked mackerel)	Legumes, nuts, and wheat (chickpeas, peanuts, cashews, bread)
Fresh vegetables and grains (carrots, squash, rice, quinoa)	Fermented soy (miso, tofu, tempeh, soy sauce)	Spices (cinnamon, chili, paprika, curry)
Mild herbs and spices (oregano, basil, ginger, mustard)	Vinegar and vinegar-based condiments	Citric-acid-containing juices
Egg yolk or cooked egg white	Raw eggs	Seafood and raw egg white

High-histamine foods contain elevated levels due to microbial fermentation or spoilage; low-histamine foods are fresh and minimally processed; histamine-releasing foods trigger endogenous histamine release or impair histamine breakdown [82,83,84,85,86].

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
