# Peer review of "Histamine Metabolism in IBD: Towards Precision Nutrition"

_nutrients, 2025, doi:10.3390/nu17152473_

Round 1

Reviewer 1 Report

Comments and Suggestions for Authors

The review entitled „Histamine Metabolism in IBD: Towards Precision Nutrition” presents basic information on the role of histamine and its metabolites in ulcerative colitis and Crohn’s disease. The abstract is clear and the discussion is interesting. The authors should add more relevancy and novelty to their figures and tables as well as include more experimental data from original papers in the subject. My detailed comments are listed below.

Figure 1. In the figure legend the shortcuts DAO and HNMT should be explained. Similarly, to all other figures.

Table 1- the authors should add relevant references associated with the histamine receptor’s role in IBD. Please do similar changes in Table 3.

Line 203-204. The sentence “The reduction of IL-10 in histamine-deficient DSS models underscores the immunomodulatory potential of mast cell-derived histamine”. The authors should add citation here. Also, additional information on the interplay between IL-2, IL-10, and histamine release in cell or animal models of IBD should be included. What mechanism is behind those changes?

Figure 2- the authors should add a graphical legend with explanation of the meaning of “orange dots”, blue and violet cells etc. The reader should be able to interpret the figure without reading the headline of the figure.

Figure 1 on page 8- should be figure 3. The authors should add more details to it and show the mechanism behind “Histamine-producing bacteria's metabolism and gut epithelial diffusion”. In the current form the figure means nothing.

Line 2070 to 273- the authors write that “Little is known about the histamine-producing taxa enriched during dysbiosis that carry the hdc operon, such as specific strains of Lactobacillus, Morganella morganii, Enterobacter cloacae, and other members of the Enterobacteriaceae family [51] (Figure 3).”. This information ends the subchapter, while it should be continued and the authors could reveal more information if there is a review of Mou et al.. The authors should refer to original data rather than review papers in this subject.

Line 306- the authors cite Table 3 in the text but it should be table 4. What this table and its content add to the novelty of the paper?

The subchapter “Role of Yeast and Salt in IBD” should contain information on the role of yeast and salt on histamine metabolism not IBD.

The review would benefit if the authors concentrate more on in vitro, in vivo studies on the role of histamine and histamine metabolism in colitis and Crohn’s disease. The study should contain the possible mechanism behind those changes. For instant the authors could discuss the papers of doi: 10.1016/j.cell.2015.10.048, doi: 10.1007/s00210-023-02565-8., doi: 10.1159/000237129. doi: 10.1038/mi.2017.121. and many others on experimental IBD.

Author Response

Comment 1: In the figure legend, the abbreviations DAO and HNMT should be explained, as should any other abbreviations in all figures.
Response 1: This comment has been addressed: all abbreviations have been added in the relevant figure and table legends, ensuring the reader can interpret them independently.

Comment 2: The authors should add relevant references regarding the role of histamine receptors in IBD, and similar additions should be made to Table 3.
Response 2: References have been added to Table 2 to support the role of histamine receptors in IBD. Similar additions have been made to Table 3 as well.

Comment 3: Lines 203–204: The sentence “The reduction of IL-10 in histamine-deficient DSS models underscores the immunomodulatory potential of mast cell-derived histamine” should be supported with a citation. Additionally, more information on the interplay between IL-2, IL-10, and histamine release in cell or animal models of IBD should be included. What mechanism explains these changes?
Response 3:: We thank the reviewer for this valuable comment. In response, we have added an appropriate citation [DOI: 10.3390/ijms24129937] to support the statement regarding the reduction of IL-10 in histamine-deficient DSS models. To address the mechanism, we have expanded the relevant section to clarify how mast cell–derived histamine modulates IL-2 and IL-10 levels through receptor-specific pathways (H₁ receptor promoting IL-2 and Th1 polarization; H₂ receptor enhancing IL-10 secretion and anti-inflammatory effects). We believe this mechanistic explanation helps clarify the immunomodulatory role of histamine in the context of IBD. Given space constraints and maintain the focus of the manuscript, we did not add further experimental details from additional cell or animal models but have ensured that the key mechanistic link is clearly described and appropriately referenced.

Comment 4: Figure 2: The authors should include a graphical legend explaining the meaning of “orange dots,” blue and violet cells, etc., so the figure can be interpreted without relying on the figure headline.
Response 4: All figures have been revised to include detailed legends, enabling readers to interpret and understand them independently.

Comment 5:Figure 1 on page 8- should be figure 3. The authors should add more details to it and show the mechanism behind “Histamine-producing bacteria's metabolism and gut epithelial diffusion”. In the current form the figure means nothing.
Response 5: Thank you for this valuable suggestion. The figure has been renumbered correctly as Figure 3. We have substantially revised the figure to include additional details illustrating the mechanism of histamine production by gut bacteria and its diffusion through the gut epithelium.

Comment 6: Line 270 to 273- the authors write that “Little is known about the histamine-producing taxa enriched during dysbiosis that carry the hdc operon, such as specific strains of Lactobacillus, Morganella morganii, Enterobacter cloacae, and other members of the Enterobacteriaceae family [51] (Figure 3).”. This information ends the subchapter, while it should be continued and the authors could reveal more information if there is a review of Mou et al. The authors should refer to original data rather than review papers in this subject.

Response 6: We appreciate the reviewer’s suggestion to expand this section and provide original data rather than relying only on reviews. In response, we have revised the section to include additional primary sources describing the regulation of the hdc operon in key histamine-producing bacteria. For instance, we now cite original studies showing that Morganella morganii is a potent histamine producer both in food and in vivo contexts (DOI: 10.1007/s000110050463), and that Lactobacillus reuteri expresses a well-characterized PLP-dependent histidine decarboxylase system (DOI: 10.1111/j.1365-2672.2012.05344). We have also incorporated metagenomic evidence (DOI: 10.1016/j.chom.2014.02.005) demonstrating the enrichment of Enterobacteriaceae with decarboxylase gene potential in IBD colonic samples. These additions provide concrete examples of taxa and mechanisms relevant to dysbiosis and histamine production, as requested.

Comment 7: Line 306- the authors cite Table 3 in the text but it should be table 4. What this table and its content add to the novelty of the paper?

The subchapter “Role of Yeast and Salt in IBD” should contain information on the role of yeast and salt on histamine metabolism not IBD.

Response 7:  We thank the reviewer for this helpful comment. In response, we have adjusted the section title to “Role of Yeast and Salt in IBD: Links to Histamine Metabolism” and added specific information on how yeast can contribute exogenous histamine through fermented foods (DOI: 10.1080/10408398.2011.582813) and stimulate mast cell degranulation and histamine release. We also clarified that high salt intake may indirectly increase histamine levels by reducing histamine-degrading Lactobacillus species and favoring histamine-producing taxa. We kept a brief context on IBD to maintain coherence and help readers understand why histamine metabolism is relevant in this disease setting. We hope this targeted update addresses the reviewer’s concern.

Comment 8: The review would benefit if the authors concentrate more on in vitro, in vivo studies on the role of histamine and histamine metabolism in colitis and Crohn’s disease. The study should contain the possible mechanism behind those changes. For instant the authors could discuss the papers of doi: 10.1016/j.cell.2015.10.048, doi: 10.1007/s00210-023-02565-8., doi: 10.1159/000237129. doi: 10.1038/mi.2017.121. and many others on experimental IBD.

Response 8: We this. We appreciate the reviewer's valuable suggestion. We have taken this constructive comment into account and aimed to include a discussion of the proposed mechanisms and relevant experimental evidence. However, there are limited original in vitro and vivo studies directly investigating the detailed links between histamine metabolism and experimental IBD, which restricts how much we can expand this part without overinterpreting sparse data. Nevertheless, we have carefully reviewed the suggested references and integrated key insights where relevant to strengthen the discussion. We hope these additions improve the mechanistic depth while maintaining the focus and scope of the review.

General Response: We thank the reviewer for his valuable comments and we hope we have responded efficiently. In the Manuscript attached, the reviewers' comments are highlighted with yellow.

Reviewer 2 Report

Comments and Suggestions for Authors

It was with great pleasure and interest that I read the manuscript entitled: “Histamine Metabolism in IBD: Towards Precision Nutrition” addressing the timely topic of dietary support for inflammatory bowel disease therapy.

The manuscript is very interesting, containing extensive coverage of key issues. It also includes an indication of limitations and future research directions.

I believe that the manuscript is of great value.

Below are comments that I consider crucial to include in the development of a revised version of the manuscript, and which I believe will help improve its quality.

  1. I find incomprehensible the insertion at the beginning of the manuscript (even before the Introduction section) of a figure that is unsigned and not referenced in the text.

I suggest giving this figure a title and inserting a reference to it in the text of the manuscript.

2 Introduction section:

- I suggest modifying the beginning of the first sentence of this section (“Inflammatory Bowel Disease (IBD) is a heterogenous term encompassing two chronic diseases that cause inflammation of the gastrointestinal tract (GIT): Crohn’s disease (CD) and ulcerative colitis (UC)”, lines 30-32). I propose: “Inflammatory bowel disease (IBD) is a term that covers a heterogeneous group of chronic diseases that cause gastrointestinal inflammation (GIT), the two main ones being Crohn's disease (CD) and ulcerative colitis (UC)”.

- In lines 86-88 it says that “However, there is currently no universally recommended diet, as nutritional needs and responses can vary widely among individuals with IBD”, while the Discussion section writes about the ESPEN guidelines: ESPEN guideline on Clinical Nutrition in inflammatory bowel disease, Bischoff, Stephan C. et al., Clinical Nutrition, Volume 42, Issue 3, 352 – 379 (pozycja 16 w sekcji References). In addition, there are published ESPGHAN guidelines: Gerasimidis K, Russell RK, Giachero F, Gkikas K, Tel B, Assa A, Bronsky J, de Ridder L, Hojsak I, Jenke A, Norsa L, Sigall-Boneh R, Sila S, Wine E, Zilbauer M, Strisciuglio C, Gasparetto M; ESPGHAN Special Interest Group in Basic and Translational Research; the ESPGHAN IBD Porto Working Group; the ESPGHAN Allied Health Professionals. Precision nutrition in pediatric IBD: A position paper from the ESPGHAN special interest group for basic science and translational research, the IBD Porto group, and allied health professionals. J Pediatr Gastroenterol Nutr. 2024 Feb;78(2):428-445. doi: 10.1002/jpn3.12096. I suggest modifying the text of the manuscript in this regard.

- At the end of the Introduction section, it is necessary to insert a clearly stated purpose of the review.

  1. I think it is necessary to supplement the text of the manuscript with the methodology of collecting and selecting the articles used in the reviewed manuscript. It is not clear what guided the authors in the selection of specific articles: was the language of publication, the year of publication, the type of publication (Original research manuscripts? Reviews? Systematic reviews?) taken into account? In the Abstract section, only a laconic sentence was inserted: “Relevant studies were identified through a comprehensive literature search in PubMed/MEDLINE, Google Scholar, ScienceDirect, Scopus, and Web of Science” (lines 13-14).
  2. Figure 1 (page 4): 1) there is no cross-reference to this figure in the text of the manuscript; 2) an explanation of abbreviations should be inserted in the footnote to the figure; 3) it is difficult to figure out what the circular area in the figure represents - that histamine degradation takes place in the small intestine? With the participation of microorganisms? This would agree for DAO, but no longer for HNMT. The figure should leave no doubt, it should be clear and readable; 4) I suggest modifying the title of the figure: “Histamine's endogenous and exogenous sources and degradation enzymes in the intestinal lumen” (instead of “Histamine's endogenous and exogenous sources and degradation enzymes in the lumen”).
  3. Table 1 (pages 4-5): in the footnote to the table, insert explanations of abbreviations.
  4. Figure 2 (page 6): 1) the various graphic elements and their descriptions on the figure are very poorly visible; 2) for what reason is the figure divided into three sections? Are these stages of some kind? If stages, what are they?; 3) inferring from the title of the figure, it is supposed to present the mechanism of altered histamine metabolism leading to histamine accumulation - unfortunately, due to the poor quality of the figure and the lack of explanations, it is difficult to read the mechanism from the figure.
  5. Figure 1 (page 8): 1) in the footnote to the figure, an explanation of the abbreviations should be inserted - HDC; 2) why is a portion of the figure title in bold? ; 3) proposed modifications to the figure: propose to remove the top block area with “Histamine Production in the gut”, propose to move the block area with “High Histamine-producing microbiota” over the top of the figure (over “Histidine → ↑Histamine”), and the block area with “Low Histamine-producing microbiota” to the middle of the figure (over “Histidine → Histamine” - only here the arrow “↓” should be added).
  6. the sentence, “Given the complex role of histamine in inflammatory processes, we suggest that IBD patients exhibiting symptoms or biomarkers indicative of histamine intolerance may benefit from a low-histamine diet” (lines 282-284), I propose to add examples of these “biomarkers.”
  7. In the sentence in lines 286-287: “A low-histamine diet has been proposed to manage symptoms caused by histamine sensitivity in IBD patients [36]” the authors inserted a reference to the literature that does not include this diet proposal. Please correct.
  8. Table 4 (page 9): 1) no explanation of what is meant by “date” inserted in several cells of the table; 2) the last row of the table in italics states: “... histamine-releasing or DAO-inhibiting foods trigger endogenous hista-mine release or impair histamine breakdown” while the last column is titled only as “Histamine Releasing”. Please modify.
  9. In addition, I suggest supplementing the text with literature references at the end of paragraphs on lines: 167-182, 197-208, 214-222, 237-249, 252-261, 326-337.
  10. References section needs to be standardized and aligned with journal guidelines.

In addition, the text needs significant improvement in terms of editorial errors and shortcomings:

  • The numbering order of both figures and tables should be improved.
  • Explanation of abbreviations should be introduced at the first place of use of a given abbreviation, there is no need to insert further explanations. Please revise this issue.
  • The abbreviation “SIBO” should be explained. - line 263.
  • In several places, literature references were inserted contrary to the journal's guidelines - round brackets were used instead of square brackets: (25) in line 131, (26) in line 134, (36) and (37) in line 138, (33) in line 148, (68) in line 375.
  • In addition, I think references (36) and (37) in line 138 were inserted incorrectly: in my opinion, they should be [26] and [27], respectively, so that the order of insertion of the references is preserved.
  • There are several literature references inserted in the text of the manuscript that are not included in the References section: (Sanat et al., 2024) at line 79, (Camus et al., 2013) at line 364, (Wilck et al., 2017) at line 378, (Guo et al., 2013; Monteleone et al., 2016; Tubbs et al., 2017) at lines 385-386, (Hamilton et al, 2015) at lines 389-390.
  • The text of the manuscript is missing a reference to the literature items [34] from the References section (Kaur, S.; Ali, A.; Siahbalaei, Y.; Ahmad, U.; Nargis, F.; Pandey, A.K.; Singh, B. Association of Diamine Oxidase (DAO) Variants with the Risk for Migraine from North Indian Population. Meta Gene 2020, 24, 100619, doi:10.1016/j.mgene.2019.100619; lines 615-616).
  • Please insert correct and standardized abbreviation: “low FODMAP”.

Author Response

Comment 1: Below are comments that I consider crucial to include in the development of a revised version of the manuscript, and which I believe will help improve its quality.

  1. I find incomprehensible the insertion at the beginning of the manuscript (even before the Introduction section) of a figure that is unsigned and not referenced in the text.

I suggest giving this figure a title and inserting a reference to it in the text of the manuscript.

Response 1: Thank you for your comment. The figure placed at the beginning of the manuscript is the graphical abstract, intended to provide a visual overview of the key concepts discussed in the review. To avoid confusion, we have now removed it from the manuscript and contacted Nutrients to insert it with a label ‘Graphical Abstract’, separating it from the main body of the manuscript.

Comment 2: Introduction section:

- I suggest modifying the beginning of the first sentence of this section (“Inflammatory Bowel Disease (IBD) is a heterogenous term encompassing two chronic diseases that cause inflammation of the gastrointestinal tract (GIT): Crohn’s disease (CD) and ulcerative colitis (UC)”, lines 30-32). I propose: “Inflammatory bowel disease (IBD) is a term that covers a heterogeneous group of chronic diseases that cause gastrointestinal inflammation (GIT), the two main ones being Crohn's disease (CD) and ulcerative colitis (UC)”.

Response 2: We appreciate the reviewer’s suggestion. In response, we have revised the beginning of the Introduction accordingly (lines 29–31) to improve clarity, incorporating the proposed phrasing. The revised first sentence of this section: “Inflammatory Bowel Disease (IBD) is a term that covers a heterogeneous group of chronic diseases that cause inflammation of the gastrointestinal tract (GIT), the two main ones being Crohn’s disease (CD) and ulcerative colitis (UC).”

Comment 3: In lines 86-88 it says that “However, there is currently no universally recommended diet, as nutritional needs and responses can vary widely among individuals with IBD”, while the Discussion section writes about the ESPEN guidelines: ESPEN guideline on Clinical Nutrition in inflammatory bowel disease, Bischoff, Stephan C. et al., Clinical Nutrition, Volume 42, Issue 3, 352 – 379 (pozycja 16 w sekcji References). In addition, there are published ESPGHAN guidelines: Gerasimidis K, Russell RK, Giachero F, Gkikas K, Tel B, Assa A, Bronsky J, de Ridder L, Hojsak I, Jenke A, Norsa L, Sigall-Boneh R, Sila S, Wine E, Zilbauer M, Strisciuglio C, Gasparetto M; ESPGHAN Special Interest Group in Basic and Translational Research; the ESPGHAN IBD Porto Working Group; the ESPGHAN Allied Health Professionals. Precision nutrition in pediatric IBD: A position paper from the ESPGHAN special interest group for basic science and translational research, the IBD Porto group, and allied health professionals. J Pediatr Gastroenterol Nutr. 2024 Feb;78(2):428-445. doi: 10.1002/jpn3.12096. I suggest modifying the text of the manuscript in this regard.

Response 3: We thank the reviewer for this valuable input. We agree that both ESPEN and ESPGHAN guidelines provide detailed, evidence-based dietary recommendations for managing IBD in adults and children. Our initial intention was to emphasize that, despite current dietary recommendations, there is still no single universally effective diet for all patients with IBD, given the heterogeneity in disease and progression of symptoms, as well as nutritional needs (e.g., Mediterranean diet vs. Low-FODMAP diet). To address this, we have revised the sentence and incorporated this valuable comment for improved clarity and to acknowledge the ESPEN and ESPGHAN guidelines while also highlighting the need for precision nutrition approaches to optimize an individual’s symptoms. The revised sentence (lines: 88-93) is: Current evidence-based dietary recommendations outlined in the clinical guidelines of ESPEN and ESPGHAN support the nutritional management of adult and pediatric IBD patients, respectively. However, However, despite these advances, further progress is still needed to develop effective precision nutrition strategies, to tailor dietary interventions to the individual needs of IBD patients, given the wide variability in nutritional requirements and responses.

Comment 4: At the end of the Introduction section, it is necessary to insert a clearly stated purpose of the review.

Response 4: We thank the reviewer for the comment. To address this, we have inserted a clearly stated purpose of the review (lines 105-108): This narrative review aims to provide an overview of histamine’s molecular signaling pathways involved in IBD and to evaluate the potential therapeutic role of a low-histamine diet in a subset of IBD patients.

Comment 5: I think it is necessary to supplement the text of the manuscript with the methodology of collecting and selecting the articles used in the reviewed manuscript. It is not clear what guided the authors in the selection of specific articles: was the language of publication, the year of publication, the type of publication (Original research manuscripts? Reviews? Systematic reviews?) taken into account? In the Abstract section, only a laconic sentence was inserted: “Relevant studies were identified through a comprehensive literature search in PubMed/MEDLINE, Google Scholar, ScienceDirect, Scopus, and Web of Science” (lines 13-14).

Response 5: We thank the reviewer for this important consideration. As this manuscript is a narrative review, a formal systematic review protocol was not followed. However, to improve transparency regarding our literature selection process, we have added a Methods section after the Introduction, outlining the databases searched, time frame, language restrictions, and types of publications considered. Additionally, we have included Supplementary Table 1, which provides the full list of keywords and search combinations used.

Comment 6: Figure 1 (page 4): 1) there is no cross-reference to this figure in the text of the manuscript; 2) an explanation of abbreviations should be inserted in the footnote to the figure; 3) it is difficult to figure out what the circular area in the figure represents - that histamine degradation takes place in the small intestine? With the participation of microorganisms? This would agree for DAO, but no longer for HNMT. The figure should leave no doubt, it should be clear and readable; 4) I suggest modifying the title of the figure: “Histamine's endogenous and exogenous sources and degradation enzymes in the intestinal lumen” (instead of “Histamine's endogenous and exogenous sources and degradation enzymes in the lumen”).

Response 5: We thank the reviewer for the detailed feedback. 1) The cross-reference to the figure has been added in the text. 2) as well as the abbreviations 3) We thank the reviewer for their valuable feedback. In response, we have revised the figure enzymes. The updated figure now explicitly differentiates between:

  • DAO, shown in the extracellular space (intestinal lumen), where it degrades luminal histamine.
  • HNMT, shown as acting intracellularly within enterocytes, is consistent with its cytosolic localization.

We have also added a legend indicating the localization (extracellular vs. intracellular), labeled the enzymes within the tissue context, and used arrows to show the direction of histamine degradation. Title has been modified accordingly.

Comment 6: Table 1 (pages 4-5): in the footnote to the table, insert explanations of abbreviations.

Response 6: We have now added explanations of all abbreviations used in Table 1 in the footnote.

Comment 7: Figure 2 (page 6): 1) the various graphic elements and their descriptions on the figure are very poorly visible; 2) for what reason is the figure divided into three sections? Are these stages of some kind? If stages, what are they?; 3) inferring from the title of the figure, it is supposed to present the mechanism of altered histamine metabolism leading to histamine accumulation - unfortunately, due to the poor quality of the figure and the lack of explanations, it is difficult to read the mechanism from the figure.

Response 7: We appreciate your comment. We have decided to remove Figure 2. It aimed to provide an overview of the different mechanisms that can lead to histamine accumulation, but we agree with you that the poor quality and complexity make it difficult to illustrate. We hope that the text and provide a clear understanding of the biological background of histamine accumulation in IBD.

Comment 8: Figure 1 (page 8): 1) in the footnote to the figure, an explanation of the abbreviations should be inserted - HDC; 2) why is a portion of the figure title in bold? ; 3) proposed modifications to the figure: propose to remove the top block area with “Histamine Production in the gut”, propose to move the block area with “High Histamine-producing microbiota” over the top of the figure (over “Histidine → ↑Histamine”), and the block area with “Low Histamine-producing microbiota” to the middle of the figure (over “Histidine → Histamine” - only here the arrow “↓” should be added).

Response 8: We thank the reviewer for this constructive feedback. Figure 1 (Now Figure 2) has been revised and edited according to the reviewers' comments. 1) The abbreviations have been added. 2) The title is edited. 3) The proposed modifications have been implemented, and we are truly grateful for the detailed instructions to provide clarity with Figure 2.

Comment 9: the sentence, “Given the complex role of histamine in inflammatory processes, we suggest that IBD patients exhibiting symptoms or biomarkers indicative of histamine intolerance may benefit from a low-histamine diet” (lines 282-284), I propose to add examples of these “biomarkers.”

Response 9: We thank the reviewer for pointing this out, and we agree that providing specific examples of biomarkers would improve the clarity and scientific value of this statement. We have revised the sentence to include examples such as reduced DAO activity and elevated histamine levels in serum, which are often considered indicative of histamine intolerance.The sentence now is: Low-histamine diet is the gold standard for HIT treatment [40,78], and given the complex role of histamine in inflammatory processes, we suggest that IBD patients exhibiting symptoms or biomarkers, such as reduced DAO activity or elevated histamine levels in serum, indicative of HIT, may benefit from its implementation.

Comment 10: In the sentence in lines 286-287: “A low-histamine diet has been proposed to manage symptoms caused by histamine sensitivity in IBD patients [36]” the authors inserted a reference to the literature that does not include this diet proposal. Please correct.

Response 10: We thank the reviewer for out, bringing this to our attention. We have now revised the sentence to clarify that the reference pertains to histamine intolerance, rather than specifically recommending the diet for IBD patients. We have also adjusted the context to accurately reflect the literature while keeping the same reference. The revised sentence now reads:Low-histamine diet is the gold standard for HIT treatment [40,78].

Comment 11: Table 4 (page 9): 1) no explanation of what is meant by “date” inserted in several cells of the table; 2) the last row of the table in italics states: “... histamine-releasing or DAO-inhibiting foods trigger endogenous histamine release or impair histamine breakdown” while the last column is titled only as “Histamine Releasing”. Please modify.

Response 11: We thank the reviewer for the detailed comment on Table 4. In response:

  1. This was a preprocessing oversight, and the repeated word “data” has now been removed from both the first and third rows of Table 4.
  2. For clarity and to avoid confusion, we have deleted DAO-inhibiting foods.

Comment 12: In addition, I suggest supplementing the text with literature references at the end of paragraphs on lines: 167-182, 197-208, 214-222, 237-249, 252-261, 326-337.

Response 12: We thank the reviewer for this valuable suggestion. We have now supplemented the text with relevant literature references at the indicated paragraph sections. However, in some instances, we have not added references at the end of paragraphs where the content highlights areas requiring further research, based on the previously cited references that address specific mechanisms. References section needs to be standardized and aligned with journal guidelines.

Comment 13: In addition, the text needs significant improvement in terms of editorial errors and shortcomings:

The numbering order of both figures and tables should be improved.

Explanation of abbreviations should be introduced at the first place of use of a given abbreviation, there is no need to insert further explanations. Please revise this issue. The abbreviation “SIBO” should be explained. - line 263.

In several places, literature references were inserted contrary to the journal's guidelines - round brackets were used instead of square brackets: (25) in line 131, (26) in line 134, (36) and (37) in line 138, (33) in line 148, (68) in line 375.

In addition, I think references (36) and (37) in line 138 were inserted incorrectly: in my opinion, they should be [26] and [27], respectively, so that the order of insertion of the references is preserved.

There are several literature references inserted in the text of the manuscript that are not included in the References section: (Sanat et al., 2024) at line 79, (Camus et al., 2013) at line 364, (Wilck et al., 2017) at line 378, (Guo et al., 2013; Monteleone et al., 2016; Tubbs et al., 2017) at lines 385-386, (Hamilton et al, 2015) at lines 389-390.

The text of the manuscript is missing a reference to the literature items [34] from the References section (Kaur, S.; Ali, A.; Siahbalaei, Y.; Ahmad, U.; Nargis, F.; Pandey, A.K.; Singh, B. Association of Diamine Oxidase (DAO) Variants with the Risk for Migraine from North Indian Population. Meta Gene 2020, 24, 100619, doi:10.1016/j.mgene.2019.100619; lines 615-616).

Please insert correct and standardized abbreviation: “low FODMAP”.

Response 13: We thank you for the detailed comments and feedback. The manuscript has been revised and corrected accordingly.

General Response: We thank the reviewer for the valuable and detailed comments. In the manuscript attached the reviewers' corrections are highlighted with blue.

Reviewer 3 Report

Comments and Suggestions for Authors

The manuscript is an elegant, thoughtful and well-executed review, which includes relevant information and is pleasant to read for the reader. I see no major content flaws to complain about and only certain formatting issues and slight additional information are necessary prior to acceptance.

Lines 214-235: In microbial dysbiois, it would be interesting if include information about the activity and relevance of gut virome, not only bacteria. You can find information at Ezzatpour et al. 2023. The Human Gut Virome and Its Relationship with Nontransmissible Chronic Diseases

Line 279: Figure 3 was wrongly named as “Figure 1”.

Table 2 is lacking in the text, or all the Tables but Table 1 are cited incorrectly in the main text.

Table 4, first raw: The work “data” was cited twice. The same in the third raw.

Section 4, “Towards precision nutrition in IBD: Low histamine diet & beyond”. The subheading of this section must be numbered.

Please number Limitations section. “future perspectives” should be “conclusions and future perspectives”.

Author Response

Comment 1: Lines 214-235: In microbial dysbiois, it would be interesting if include information about the activity and relevance of gut virome, not only bacteria. You can find information at Ezzatpour et al. 2023. The Human Gut Virome and Its Relationship with Nontransmissible Chronic Diseases

Response 1: We thank the reviewer for this interesting suggestion and for directing us to the recent work by Ezzatpour et al. (2023). Indeed, the gut virome represents an emerging and exciting area of research in IBD and chronic disease. However, in the present narrative review, we have chosen to focus on the bacterial component of the gut microbiota due to its better-established role in histamine production and metabolism, with a few human studies, which is central to our discussion. The contribution of the gut virome to histamine metabolism remains largely unexplored, and its potential modulation through diet is still under investigation. Due to time constraints and our focus on precision nutrition approaches, we have not included an in-depth discussion of the virome in this version. However, we acknowledge its potential relevance and have now briefly mentioned it in the Future Perspectives section, following the gut microbiome, as a direction for future research.

Comment 2: Line 279: Figure 3 was wrongly named as “Figure 1”.

Response 2:  We thank you for the comment, it is now named correctly.

Comment 3: Table 2 is lacking in the text, or all the Tables but Table 1 are cited incorrectly in the main text.

Response 3: We thank the reviewer for out, bringing this to our attention. All table citations have now been carefully reviewed and corrected. Table 2, along with the other tables, is now properly cited in the main text.

Comment 4: Table 4, first raw: The work “data” was cited twice. The same in the third raw.

Response 4: We thank the reviewer for bringing this to our attention. This was a preprocessing oversight, and the repeated word “data” has now been removed from both the first and third rows of Table 4.

Comment 5: Section 4, “Towards precision nutrition in IBD: Low histamine diet & beyond”. The subheading of this section must be numbered.

Response 5: We thank the reviewer for out, bringing this to our attention. We have now updated the manuscript to ensure that all headings and subheadings, including Section 4 (“Towards precision nutrition in IBD: Low histamine diet & beyond”), are properly numbered for consistency and clarity.

Comment 6: Please number Limitations section. “future perspectives” should be “conclusions and future perspectives”.

Response 6: Thank you for the comment. We have also now numbered the Limitations section accordingly.

General comment: We thank the reviewer for the comments. In the attached manuscript, the reviewers' comments are highlighted in green.

Round 2

Reviewer 1 Report

Comments and Suggestions for Authors

The authors answered all my comments and improved the manuscript.

Reviewer 2 Report

Comments and Suggestions for Authors

I would like to thank the authors for reading my comments and for taking them into account in preparing the revised version of the manuscript.